# Ductile Copolyesters Prepared Using Succinic Acid, 1,4-Butanediol, and Bis(2-hydroxyethyl) Terephthalate with Minimizing Generation of Tetrahydrofuran

**DOI:** 10.3390/polym16040519

**Published:** 2024-02-14

**Authors:** Sang Uk Park, Hyeon Jeong Seo, Yeong Hyun Seo, Ju Yong Park, Hyunjin Kim, Woo Yeon Cho, Pyung Cheon Lee, Bun Yeoul Lee

**Affiliations:** Department of Molecular Science and Technology, Ajou University, Suwon 16499, Republic of Korea; gkdl3094@ajou.ac.kr (S.U.P.); hjseo@ajou.ac.kr (H.J.S.); tdg0730@ajou.ac.kr (Y.H.S.); pjy3741@ajou.ac.kr (J.Y.P.); khj610@ajou.ac.kr (H.K.); wycho418@ajou.ac.kr (W.Y.C.); pclee@ajou.ac.kr (P.C.L.)

**Keywords:** PBS, poly(1,4-butylene succinate), bis(2-hydroxyethyl) terephthalate, copolyester, polycondensation, biodegradable

## Abstract

Poly(1,4-butylene succinate) (PBS) is a promising sustainable and biodegradable synthetic polyester. In this study, we synthesized PBS-based copolyesters by incorporating 5–20 mol% of –O_2_CC_6_H_4_CO_2_– and –OCH_2_CH_2_O– units through the polycondensation of succinic acid (SA) with 1,4-butanediol (BD) and bis(2-hydroxyethyl) terephthalate (BHET). Two different catalysts, H_3_PO_4_ and the conventional catalyst (nBuO)_4_Ti, were used comparatively in the synthesis process. The copolyesters produced using the former were treated with M(2-ethylhexanoate)_2_ (M = Mg, Zn, Mn) to connect the chains through ionic interactions between M^2+^ ions and either –CH_2_OP(O)(OH)O^−^ or (–CH_2_O)_2_P(O)O^−^ groups. By incorporating BHET units (i.e., –O_2_CC_6_H_4_CO_2_– and –OCH_2_CH_2_O–), the resulting copolyesters exhibited improved ductile properties with enhanced elongation at break, albeit with reduced tensile strength. The copolyesters prepared with H_3_PO_4_/M(2-ethylhexanoate)_2_ displayed a less random distribution of –O_2_CC_6_H_4_CO_2_– and –OCH_2_CH_2_O– units, leading to a faster crystallization rate, higher *T*_m_ value, and higher yield strength compared to those prepared with (nBuO)_4_Ti using the same amount of BHET. Furthermore, they displayed substantial shear-thinning behavior in their rheological properties due to the presence of long-chain branches of (–CH_2_O)_3_P=O units. Unfortunately, the copolyesters prepared with H_3_PO_4_/M(2-ethylhexanoate)_2_, and hence containing M^2+^, –CH_2_OP(O)(OH)O^−^, (–CH_2_O)_2_P(O)O^−^ groups, did not exhibit enhanced biodegradability under ambient soil conditions.

## 1. Introduction

Polymers play an indispensable role in modern civilization, with annual production reaching approximately 400 million tons. However, their widespread use in various applications, including plastics, fibers, elastomers, coatings, and adhesives, is currently met with significant opposition due to concerns related to post-consumer waste and sustainability [1]. A staggering 30% of produced plastic waste is disposed of improperly, posing serious environmental challenges, and most of it is produced using fossil-based resources [2]. One promising approach to address these concerns is through closed-loop recycling of post-consumer polymers, often referred to as a circular economy [3,4,5,6,7,8,9]. However, in certain sectors, such as agriculture, recycling can be inherently challenging, making the promotion or enforcement of biodegradable polymers a viable alternative [10,11,12]. In this context, various types of biodegradable polymers have been explored [13,14,15,16,17,18,19,20,21,22,23,24,25,26,27,28]. While cellulose and starch-based polymers remain significant sources of biodegradable materials, there has been a recent surge in the production capacity of biodegradable synthetic polyesters like PBAT [poly(butylene adipate-*co*-terephthalate)] and PLA (polylactide) [29,30]. PLA, for instance, is a sustainable polymer derived from renewable resources. However, it only composts under harsh artificial composting conditions and does not biodegrade at ambient conditions, limiting its acceptance in biodegradable polymer markets [2]. On the other hand, PBAT exhibits somewhat better biodegradability than PLA, with certain grades even receiving biodegradable certifications at ambient conditions [2]. However, it is important to note that raw materials used in PBAT production, particularly terephthalic acid and adipic acid, are currently derived from fossil-based sources.

PBS stands as another promising sustainable and biodegradable synthetic polyester that has garnered significant attention from numerous companies aiming for commercialization [31,32,33]. However, its path to commercial success has not been as smooth as that of PLA and PBAT, and its accessibility remains somewhat limited. The primary obstacle to its widespread adoption may be attributed to the cost of the raw materials required for PBS production, namely, SA and BD. Currently, bio-based SA, a critical ingredient in PBS production, tends to be relatively more expensive compared to the fossil-based terephthalic acid and adipic acid utilized in the PBAT production. Nevertheless, there is mounting pressure to shift towards the production of bio-based polymers using monomers sourced from renewable resources [34]. This growing impetus has spurred considerable efforts towards the production of SA and BD through fermentation processes, leading to successful commercial production initiatives that have been announced and are currently expanding [35,36,37].

PBS is a semi-crystalline polymer known for its melting temperature (*T*_m_) of approximately 115 °C. It has traditionally been characterized as a brittle polymer with a high tensile strength ranging from 20 to 30 MPa but a low elongation at break, typically below 100% [38,39,40], even though there are reports indicating significantly higher tensile strengths of 59 and 42 MPa accompanied by excellent elongation at break values of 700 and 230% [41,42]. In pursuit of either altering its mechanical properties or reducing production costs, researchers have synthesized various derivatives of PBS such as poly(butylene succinate-*co*-adipate) (PBSA) [43,44], poly(butylene succinate-*co*-terephthalate) (PBST) [40,45,46], poly(butylene succinate-*co*-furandicarboxylate) [47], poly(butylene succinate-*co*-2-methylsuccinate) [39], poly(butylene succinate-*co*-isosorbide succinate) [24], poly(butylene succinate-*co*-ethylene succinate) [41,48], poly(butylene succinate-*co*-caprolactone) [49], PBS-*block*-poly(butylene terephthalate) [50], PBS-*block*-polydimethylsiloxane [51], and PBS containing side chain onium salts [52], providing insights into their distinctive properties.

Typically, PBS and its derivatives have been synthesized by (RO)_4_Ti-catalyzed diacid/diol polycondensation. However, the molecular weights attainable by the polycondensation are reported to be limited, *M*_w_ < 100 kDa, due to simultaneous occurrence of competing reactions of polycondensation and degradation at the latter stage [53,54], and chain extension reactions have been performed at the final stage of polycondensation [55,56]. Many articles suggested the use of a diisocyanate such as 1,6-hexamethylene diisocyanate (HDI) as a chain-extending agent. However, the disadvantage is that the chain extender integration tends to reduce biodegradability [53,57]. Recently, we have pioneered a novel approach for the synthesis of rapidly biodegradable polyesters [58]. In this method, chain extension is achieved through ionic bonds formed between phosphate groups [(~OP(O)(OH)O^−^ or (~O)_2_P(O)O^−^] and divalent metal ions, such as Zn^2+^ or Mg^2+^. The process involves the facile synthesis of polyesters containing phosphate groups [~OPO(OH)_2_ or (~O)_2_P(O)OH] by simply substituting the (RO)_4_Ti catalyst with H_3_PO_4_ during the polycondensation. These resulting PBATs exhibit significantly enhanced biodegradability, making them potentially useful as slow-releasing fertilizers during composting due to their content of plant growth nutrients—phosphate and metal ions [59,60].

In this study, we have extended our innovative approach by synthesizing PBS derivatives. Specifically, we carried out the SA/BD polycondensation process in the presence of an additional compound, BHET, with the aim of imparting ductile properties and evaluating biodegradability (Figure 1). In a prior communication [58], it was demonstrated that the ionic aggregates of PBS, where chains are elongated with phosphate groups and divalent metal ions, displayed brittle characteristics with an elongation at break of only 20%, and its biodegradability was not assessed. BHET is recognized as a crucial intermediate in the closed-loop recycling of poly(ethylene terephthalate) (PET). Waste PET is converted into BHET, which can then be transformed back into PET, making it a potentially abundant and cost-effective chemical resource for the near future [61,62,63]. Incorporating terephthalate units into aliphatic polyesters or aliphatic polycarbonates is typically crucial in enhancing their thermal and mechanical properties [64]. However, the direct utilization of terephthalic acid or dimethyl terephthalate in the polycondensation process presents challenges due to their intractability or sublimation tendencies. The use of BHET offers a solution to circumvent these issues.

## 2. Materials and Methods

### 2.1. General Remarks

Phosphoric acid (85.0%), SA (≥99.0%), (nBuO)_4_Ti (97%), and Mg(OH)_2_ were purchased from Sigma-Aldrich (St. Louis, MO, USA). BD (99.5%, 18 L scale) was purchased from Duksan (Reagents Duksan, Ansan, Korea). 2-Ethylhexanoic acid was purchased from TCI (Tokyo, Japan). Zn(2-ethylhexanoate)_2_ (80 wt% in mineral spirits), and Mn(2-ethylhexanoate)_2_ (40 wt% in mineral spirits) were purchased from Alfa Aesar (Ward Hill, MA, USA). ^1^H NMR (600 MHz), ^13^C NMR (150 MHz), and ^31^P NMR (243 MHz) analyses were performed by using a JEOL ECZ 600 instrument (JEOL, Tokyo, Japan). The GPC data were obtained in CHCl_3_ at 40 °C using a Waters Millennium apparatus with polystyrene standards. The *T*_m_, *T*_g_, *T*_c_, *T*_cc_ and their enthalpy (Δ*H*) data were determined with DSC 200 F3 Maia (NETZSCH, Bavaria, Germany) at a heating rate of 10 °C/min. For tensile studies, the polymer samples were compressed between hot plates at 130 °C. The pressure was applied incrementally, starting from 1 MPa to 5 MPa, with each 1 MPa increment lasting 5 min, totaling 20 min. Subsequently, the pressure was increased from 5 MPa to 10 MPa, with each 1 MPa increment lasting 10 min, accumulating to a total duration of 60 min. Tensile tests were performed according to ASTM D882 [65] using a Instron 3367 UTM (Instron, MA, USA) at a drawing rate of 50 mm/min with a gauge length of 50 mm.

### 2.2. Preparation of Mg(2-Ethylhexanoate)_2_

Mg(OH)_2_ (39.5 g, 0.644 mol) and 2-ethylhexanoic acid (371 g, 2.57 mol, 4.0 equivalents) were placed in a 500 mL reaction vessel equipped with a mechanical stirrer. The reactor was connected to a manifold and evacuated for 30 min. After filling it with N_2_ gas, the heating mantle’s temperature was set to 80 °C. All solids dissolved within 4 h. The temperature was then reduced to 60 °C, and the generated water was removed by evacuation (0.3 mmHg) for 12 h. This process yielded a viscous oil (193 g, constituting 50 wt% in 2-ethylhexanoic acid), which was used as obtained.

### 2.3. Preparation of PBS[5.0T; 1.0P; 210; Mg]

BD (90.1 g, 1.00 mol), SA (118 g, 1.00 mol), and BHET (12.7 g, 50 mmol, 5.0 mol% per succinic acid) were placed into a 500 mL reaction vessel equipped with a mechanical stirrer and a distillation setup. After a 30 min evacuation, the reactor was purged with N_2_ gas. The esterification reaction was initiated by heating to 190 °C using the heating mantle and was allowed to proceed for 3 h, during which water was continuously removed through the distillation setup equipped with a receiver. To collect any distillates, the receiver was cooled using a dry ice/acetone bath, while the pressure was equalized to atmospheric pressure through a mercury bubbler. H_3_PO_4_ (85 wt%, 1.15 g, 10 mmol) was then added using a syringe. Subsequently, the receiver flask (250 mL) used for collecting water during the esterification reaction was replaced with an empty one (100 mL) under a nitrogen flux. While maintaining the temperature at 190 °C, the evacuation level was gradually increased over 1 h until full evacuation was achieved (0.3–0.4 mbar). Next, the heating mantle temperature was raised to 210 °C, and polycondensation was carried out under full evacuation for 4 h. As the molecular weight increased, the stirring rate of our mechanical stirrer was automatically decreased from 250 to 60 rpm. The reactor was purged with N_2_ gas, and Mg(2-ethylhexanoate)_2_ was introduced into the reactor using a syringe. Volatile components (2-ethylhexanoic acid) generated during this step were removed through full re-evacuation at a mantle temperature of 210 °C for 1 h, eventually reaching a stirring rate of 1–3 rpm. The polymer melt was extracted from the reactor while it was still hot, and it was subsequently cut into pieces using a paper cutter before being allowed to fully crystallize.

### 2.4. Preparation of PBS[5.0T; Ti; 230]

The esterification process was conducted using the same equipment and chemicals as those used in the preparation of PBS[5.0T; 1.0P; 210; Mg]. After introducing (nBuO)_4_Ti (0.24 g, 0.72 mmol) with a syringe, the evacuation level was gradually increased at 190 °C over the course of 1 h until full evacuation was achieved (0.3–0.4 mbar). Subsequently, polycondensation was carried out at a mantle temperature of 230 °C for 4 h, finally reaching a state where the operation of the mechanical stirrer was halted. Finally, H_3_PO_4_ (85 wt%, 0.080 g, 0.82 mmol, 1.15 equivalents per (nBuO)_4_Ti) was introduced using a syringe to deactivate the Ti catalyst, following a purge with N_2_ gas. The resulting polymer melt was handled in the same manner as the preparation of PBS[5.0T; 1.0P; 210; Mg].

### 2.5. Biodegradation Studies

Soil samples were collected from three locations in South Korea for biodegradability assays: Mt. Gwanggyosan, Suwon (37°20.1360 N 127°1.1640′ E); Ajou University’s field campus, Suwon (37°17.1210 N 127°2.6710 E); and Hwaseong flower beds (37°10.8030 N 126°58.8720 E), following ISO 17556 [66] guidelines. The soil was air-dried, homogenized, and sieved (2 mm mesh). PBS[5.0T; 1.0P; 210; M] (M = Mg, Zn, Mn) were ground and sieved (2 mm mesh) using a Type ZM 200 Ultra Centrifugal Mill (Retsch, Haan, Germany). Each reactor contains a mixture of 5 g PBS derivative, 800 g soil, 200 mL distilled water, and nutrients (0.2 g KH_2_PO_4_, 0.1 g MgSO_4_, 0.4 g NaNO_3_, 0.4 g CO(NH_2_)_2_, 0.4 g NH_4_Cl per kg soil), incubated aerobically at 25 °C in the dark using a 12-channel ECHO^®^ respirometer (ECHO Instruments, Slovenske Konjice, Slovenia). Microcrystalline cellulose (20 μm, Sigma-Aldrich, USA) was used as a control, prepared under similar conditions. Blank soil samples were also prepared to normalize the evolved CO_2_. CO_2_ evolution within the reactors was continuously monitored in real-time using a near-infrared (NIR) detector in the ECHO^®^ system. The system maintained an airflow rate of 200 mL/min, which was regulated by an integrated mass flow controller.

## 3. Results and Discussion

### 3.1. SA/(BD + BHET) Polycondensation

Conventionally, the synthesis of PBS involves a two-step process. The first step is the esterification of SA and BD, which takes place at atmospheric pressure with continuous removal of generated H_2_O without the need for an additional catalyst. This step results in the formation of an oligomer containing –OH end groups. The second step involves a transesterification reaction with the removal of the generated BD under evacuation conditions. Typically, titanium alkoxide [(RO)_4_Ti] is used as the catalyst in this step [53]. To ensure the formation of oligomers containing –OH end groups, an excess of BD is added relative to SA. In our experiments, we performed the SA/BD esterification reaction by introducing additional BHET at 190 °C for 3 h with removal of any generated water. The quantity of BHET added was varied at 5, 10, 15, and 20 mol% per SA, while adjusting the amount of BD to maintain an [–OH]/[–CO_2_H] ratio of 1.05. Subsequently, we carried out the polycondensation with feeding H_3_PO_4_ (1.0, 2.0, or 3.0 mol% per SA), replacing the conventional (RO)_4_Ti catalyst, and with gradually increasing the evacuation level over 1.0 h, ultimately reaching a full vacuum state. The polycondensation process was then continued under full evacuation, with the removal of the generated BD or, if present, ethylene glycol, at elevated temperatures of 200, 210, or 220 °C to obtain thick and viscous polymer melts.

The ^31^P NMR spectra clearly indicate that H_3_PO_4_ not only acted as a catalyst during the polycondensation process but was also integrated into the resulting polyester chains, leading to the formation of distinct chemical groups. These groups include chain-end-positioned monoester –CH_2_OP(O)(OH)_2_ observed at approximately 2.5 ppm, inner-chain-positioned diester (–CH_2_O)_2_P(O)OH at around 1.5 ppm, and branching point triester (–CH_2_O)_3_P=O at approximately –0.5 ppm (Figure 1). The first two groups may not only act as catalysts due to their acidity, which is comparable to that of H_3_PO_4_, but may also undergo conversion to –OP(O)(OH)O^−^ and (–O)_2_P(O)O^−^ while being charge-balanced by M^2+^ ions. This conversion process can result in the connection of polymer chains through ionic interactions. The relative abundance of these three units varies depending on several factors such as reaction temperature, the quantity of BHET used in the feed, and the amount of H_3_PO_4_ added. For example, when preparing the copolyester at a lower temperature of 190 °C using 5.0 mol% BHET and 1.0 mol% H_3_PO_4_, the copolyester obtained at this conditions being denoted as PBS[5.0T; 1.0P; 190], the mole ratio of monoester/diester/triester, as determined from the intensity of the ^31^P NMR signals, was roughly 17:51:32. However, it is worth noting that these ratios were semi-quantitative due to a significant amount of noise in the baseline, even with extended data-acquisition times. This suggests that the inner-chain-positioned diester is the predominant group under these conditions. As the polycondensation temperature is increased to 200, 210, and 220 °C while keeping the BHET and H_3_PO_4_ feed amounts constant at 5.0 and 1.0 mol%, respectively, the ratios gradually shift to 13:45:42, 8:44:48, and 0:30:70, respectively (Figure 1a). Namely, at 220 °C, approximately 70% of the initially added H_3_PO_4_ transforms into the triester group, resulting in polymer chains with long-chain branches, while the remaining 30% forms the inner-chain-positioned diester. When the quantity of BHET in the feed is increased, the contents of monoester and diester units decrease significantly (Figure 1b). For instance, PBS[0T; 1.0P; 210], which does not contain any terephthalate units, exhibits a ratio of 25:58:17 for monoester/diester/triester, indicating substantial amounts of monoester and diester units. However, this ratio decreases to 8:44:48, 4:36:60, and 2:28:70 for PBS[5.0T; 1.0P; 210], PBS[10T; 1.0P; 210], and PBS[15T; 1.0P; 210], respectively, suggesting that chain-end-positioned monoester units are almost negligible. Notably, the ratios do not show significant changes with an increase in H_3_PO_4_ feed, remaining relatively constant at 8:44:48, 10:50:40, and 17:51:32 for PBS[5.0T; 1.0P; 210], PBS[5.0T; 2.0P; 210], and PBS[5.0T; 3.0P; 210], respectively (Appendix A).

In the ^1^H NMR spectra of the resulting copolyesters, we observed signals corresponding to terephthalate (–O_2_CC_6_*H*_4_CO_2_–, referred to as T) and succinate (–O_2_CC*H*_2_C*H*_2_CO_2_–, referred to as S) at 8.1 and 2.6 ppm, respectively. The intensities of these signals precisely matched the [BHET]/[SA] feed ratio. Additionally, a signal corresponding to the inner CH_2_ units in BD (i.e., –OCH_2_C*H*_2_C*H*_2_CH_2_O–) was observed at 1.7 ppm as a prominent signal flanked by two low-intensity signals. However, the signals related to –C*H*_2_OC(O)– were more complex, with a major signal at 4.1 ppm and many low-intensity signals in the downfield region between 4.1 and 4.7 ppm. Importantly, these low-intensity signals were well-resolved, and each of them could be definitively assigned (Figure 2) [67]. Upon completing the signal assignments, it became apparent that nearly all of the –OCH_2_CH_2_O– units initially present in BHET remained incorporated in the resulting copolyesters, primarily attaching to terephthalate units. For example, the intensity ratio of ([TOC*H*_2_C*H*_2_OT] (referred to as TET, 4.66 ppm) + [TOC*H*_2_C*H*_2_OS] (TES, 4.50 and 4.42 ppm) + [SOC*H*_2_C*H*_2_OS] (SES, 4.25 ppm))/[T] was 1.9 for PBS[5.0T; 1.0P; 210], indicating that 95% of the –OCH_2_CH_2_O– units initially present in BHET remained in the copolyester chains as incorporated. Similarly, the intensity ratio of ([TET] + [TES])/([TET] + [TES] + [SES]) was 0.70, suggesting that 70% of the incorporated –OCH_2_CH_2_O– units remained attached to terephthalate units. These ([TET] + [TES]) + [SES])/[T] ratios consistently fell within the range of 1.8–1.9, and the ([TET] + [TES])/([TET] + [TES] + [SES]) ratios remained steady at 0.64–0.72 throughout all experiments performed with variations in the feed quantity of BHET, H_3_PO_4_ and temperature (Table 1).

A relatively low-molecular-weight copolyester, PBS[5.0T; 1.0P; 190], was obtained with a reasonably narrow molecular-weight distribution at a transesterification temperature of 190 °C (*M*_w_ = 33 kDa; Ð (dispersity) = *M*_w_/*M*_n_ = 2.1). When the temperature was increased to 200 and 210 °C, the *M*_w_ values showed a marginal increase to 45 kDa, along with a slight increase in the Ð value to 2.4. However, raising the temperature further to 220 °C resulted in doubling the *M*_w_ value to 81 kDa, although the *M*_n_ value marginally increased from 20 to 26 kDa, consequently leading to a substantial increase in the Ð value from 2.4 to 3.1 (Appendix A). In the ^31^P NMR spectrum of PBS[5.0T; 1.0P; 220], the monoester –CH_2_OP(O)(OH)_2_ units were absent, while the branching-point-triester (–CH_2_O)_3_P=O units were predominant (70 mol%; Figure 1a). This suggests the formation of copolyester chains containing a substantial number of long-chain branches, which might be the cause of the increase in *M*_w_ and Ð values. Increasing the BHET feed quantity also resulted in higher *M*_w_ and Ð values, and PBS[5.0T; 1.0P; 210], PBS[10T; 1.0P; 210], and PBS[15T; 1.0P; 210] exhibited *M*_w_ values of 44, 62, and 70 kDa, respectively, with Ð values of 2.3, 2.9, and 3.1, respectively (Appendix A). On the other hand, increasing the H_3_PO_4_ feed quantity from 1.0 to 2.0 mol% had only a marginal influence on *M*_w_ and Ð values, and PBS[5.0T; 1.0P; 210] and PBS[5.0T; 2.0P; 210] showed *M*_w_ values of 44 and 47 kDa, respectively, with Ð values of 2.3 and 2.5, respectively (Appendix A). However, copolyester prepared with 3.0 mol% H_3_PO_4_, i.e., PBS[5.0T; 3.0P; 210], was not freely soluble in CHCl_3_, which hindered filtration for GPC analysis. This suggests the possibility of gel formation when a high amount of H_3_PO_4_ is used.

Analysis of the ^1^H NMR spectrum of the volatiles collected during the esterification reaction, conducted with 1.00 mole of SA, 1.00 mole of BD, and 0.050 mole of BHET, revealed the formation of 1.5 moles of H_2_O, along with a small amount of THF (11 mmol). This observation suggests that the esterification reaction, which generates H_2_O, was not fully completed, with substantial amounts of –CO_2_H and –OH groups still remaining, while the formation of the side product THF was minimal (1.1 mol% per BD). The significant formation of THF presented a considerable challenge during the polycondensation of terephthalic acid and BD. For instance, the use of terephthalic acid is not feasible for producing poly(butylene terephthalate) due to the substantial generation of THF. To address this problem, dimethyl terephthalate is employed as an alternative, even though it does not completely eliminate the formation of THF [68]. When attempting esterification using 5.0 mol% dimethyl terephthalate and 10 mol% ethylene glycol as a direct substitute for 5.0 mol% BHET, a significant issue arose in our experiments. Dimethyl terephthalate underwent sublimation, adhering to the walls of the reactor and the glass lines, which led to obstruction of the reaction (Appendix A). It is worth mentioning that this fouling of the reactor was not encountered when the esterification reaction was conducted using BHET, highlighting the inherent advantage of utilizing BHET over the conventional approach of using terephthalic acid or dimethyl terephthalate (Appendix A). Indeed, the preparation of copolyesters comprising –OCH_2_CH_2_CH_2_CH_2_O–, –OCH_2_CH_2_O–, succinate, and terephthalate units has been reported [67]. However, it was not directly synthesized using BD, ethylene glycol, SA, and terephthalic acid. Instead, it was prepared from prepolymers of 1,4-butylene succinate, ethylene succinate, and ethylene terephthalate.

The prepared copolyesters, which contained acidic –CH_2_OP(O)(OH)_2_ and (–CH_2_O)_2_P(O)OH groups, exhibited instability and a severe deterioration in their mechanical strength during storage. To address this issue, the copolyesters were treated with divalent metal 2-ethylhexanoate [(CH_3_(CH_2_)_2_CH(Et)CH_2_CO_2_)_2_M; M = Mg, Zn, Mn; 1.0 eq/H_3_PO_4_]. It is important to note that divalent Mg, Mn, and Zn ions are essential nutrients in plant growth. Considering the pKa values of the 2-ethylhexanoate anion, nBuOP(O)(OH)_2_, and (nBuO)_2_P(O)(OH), which are 4.8, 1.8, and 1.7, respectively, the –CH_2_OP(O)(OH)_2_ and (–CH_2_O)_2_P(O)OH units underwent conversion to –OP(O)(OH)O^−^ and (–O)_2_P(O)O^−^ while being charge-balanced by M^2+^ ions upon the addition of the metal 2-ethylhexanoate. This conversion process consequently resulted in the connection of polymer chains through ionic interactions. This conversion process also led to the simultaneous generation of 2-ethylhexanoic acid. Upon the removal of the generated 2-ethylhexanoic acid, the viscosity of the polymer melts significantly increased, leading to a state where the mechanical stirrer in our polymerization setup ceased.

For comparison, we synthesized copolyesters using the conventional method, employing (nBuO)_4_Ti catalyst (200 ppm-Ti/product) and varying the quantities of BHET in the feed (entries 10–13). Surprisingly, under our polymerization conditions and settings, we achieved significantly high-molecular-weight PBS and copolyesters. Molecular weight (*M*_w_) values of approximately 300 kDa were obtained for feed ratios of 0, 5.0, and 10 mol% BHET per SA, whereas *M*_w_ reached 220 kDa for a feed of 15 mol% BHET. The ratios of [TET]:[TES]:[SES]:[TBT]:[TBS]:[SBS] measured in ^1^H NMR spectra were notably different from those observed for copolyesters prepared using H_3_PO_4_. Extensive transesterification reactions occurred, possibly due to the use of more reactive (nBuO)_4_Ti catalyst or the elevated polymerization temperature of 230 °C. This resulted in the appearance of signals related to TBT units in the ^1^H NMR spectra, albeit with relatively low intensities, which were absent in the copolyesters prepared using H_3_PO_4_ (Figure 2). In addition to this observation, signals associated with TES units decreased while those corresponding to TBS units increased. Consequently, the intensity ratio of ([TET] + [TES])/([TET] + [TES] + [SES]) decreased from 0.64–0.72 to 0.20–0.41 when H_3_PO_4_ was replaced with (nBuO)_4_Ti catalyst.

### 3.2. Thermal Properties

In differential scanning calorimetry (DSC) studies, PBS[0T; 1.0P; 210; Mg], which refers to copolyesters obtained by treating PBS[0T; 1.0P; 210] with magnesium 2-ethylhexanoate, exhibited nearly identical thermal properties to PBS[0T; Ti; 230] prepared through the conventional method using (nBuO)_4_Ti catalyst, (*T*_m_ (melting temperature), 116 and 114 °C; *T*_c_ (crystallization temperature), 68 and 65 °C; *T*_g_ (glass transition temperature), –34 and –34 °C, respectively). Usually, thermal properties do not exhibit significant sensitivity to variations in end groups or the attachment of a side group. Upon the incorporation of BHET units (specifically, –O_2_CC_6_H_4_CO_2_– and –OCH_2_CH_2_O– units), the *T*_m_ value decreased to 106 °C for PBS[5.0T; 1.0P; 210; Mg] and further decreased gradually with increasing BHET feed quantities, PBS[10T; 1.0P; 210; Mg] and PBS[15T; 1.0P; 210; Mg] exhibiting *T*_m_ signals at 96 and 87 °C, respectively (Figure 3). The copolyester obtained by feeding 20 mol% BHET, i.e., PBS[20T; 1.0P; 210; Mg], showed no *T*_m_ signal during its second heating scan, although it exhibited a clear *T*_m_ at 77 °C during the first heating scan. This observation suggests that PBS[20T; 1.0P; 210; Mg] is crystalline, but its crystallization rate is significantly slow. The signal intensities, specifically the heat of melting (Δ*H*), were substantially high for PBS[0T; 1.0P; 210; Mg], PBS[5.0T; 1.0P; 210; Mg], and PBS[10T; 1.0P; 210; Mg], with Δ*H* values of 61, 55, and 47 J/g, respectively. In contrast, the Δ*H* value was relatively weak for PBS[15T; 1.0P; 210; Mg], measuring only 18 J/g.

The *T*_c_ signals, observed during the cooling process after the first heating scan, were significant at 68 and 48 °C, accompanied by Δ*H* values of 67 and 55 J/g, for PBS[0T; 1.0P; 210; Mg] and PBS[5.0T; 1.0P; 210; Mg], respectively, while the *T*_cc_ signals, observed during the second heating process, were feeble. These observations suggest that the crystallization process occurred rapidly for PBS and the copolyester prepared with a small amount of BHET (5 mol%). Interestingly, PBS[10T; 1.0P; 210; Mg] exhibited both *T*_c_ and *T*_cc_ signals at 28 and 19 °C, respectively, with substantial intensities. This indicates the occurrence of the crystallization process at room temperature at a relatively slow pace. In contrast, for PBS[15T; 1.0P; 210; Mg], the *T*_c_ signal was not apparent, and it only exhibited a significant *T*_cc_ signal over a broad temperature range of 20–70 °C, indicating a slow crystallization rate. As for PBS[20T; 1.0P; 210; Mg], neither *T*_c_ nor *T*_cc_ signals were observed.

The thermal properties remained almost unchanged despite variations in polycondensation temperature and the type of metal ions used at the final stage (Table 2, entries 6–8). However, due to a more random distribution of –O_2_CC_6_H_4_CO_2_– and –OCH_2_CH_2_O– units, copolyesters synthesized with (nBuO)_4_Ti exhibited a more significant reduction in *T*_m_ and its associated Δ*H* value when exposed to the same amount of BHET feed. For example, PBS[5.0T; Ti; 230] and PBS[10T; Ti; 230] displayed *T*_m_ signals at 99 and 86 °C, with Δ*H* values of 47 and 33 J/g, respectively. In contrast, PBS[5.0T; 1.0P; 210; Mg] and PBS[10T; 1.0P; 210; Mg] exhibited *T*_m_ signals at 106 and 96 °C, with Δ*H* values of 55 and 47 J/g, respectively. Both PBS[15T; Ti; 230] and PBS[20T; Ti; 230] did not exhibit any *T*_m_ signals during the second heating process. Additionally, PBS[5.0T; Ti; 230] displayed a *T*_c_ signal over a broad temperature range of 15–60 °C, while PBS[10T; Ti; 230] only showed a broad *T*_cc_ signal at 15–50 °C, without the appearance of a *T*_c_ signal. The slow crystallization rate is a significant issue in copolyesters prepared using prepolymers of 1,4-butylene succinate, ethylene succinate, and ethylene terephthalate [67].

The glass transition temperatures (*T*_g_) of PBS[0T; 1.0P; 210; Mg] and PBS[0T; Ti; 230] measured on DSC curves were both –34 °C, and they showed a slight increase upon the incorporation of BHET units, with both PBS[*x*T; 1.0P; 210; Mg] and PBS[*x*T; Ti; 230] (*x* = 5.0, 10, 15, 20) series showing *T*_g_ signals ranging from –30 °C to –24 °C, with higher *T*_g_ values corresponding to a higher BHET content. This same systematic trend was also evident in dynamic mechanical analysis (DMA). The *T*_g_ values measured on storage modulus, loss modulus, and tanδ curves (i.e., E′_onset_, E″_max_, tanδ_max_) for PBS[0T; 1.0P; 210; Mg] were –30, –20, and –14 °C, respectively, which were notably higher than that measured using DSC (–34 °C). These values systematically increased with the increase of BHET content, reaching –16, –5.3, and 4.0 °C, respectively, for PBS[20T; 1.0P; 210; Mg] (Figure 4). A similar trend was also observed for the PBS[*x*T; Ti; 230] (*x* = 0, 5.0, 10, 15, 20) series (Appendix A), although these series displayed slightly lower *T*_g_ values compared to their corresponding PBS[*x*T; 1.0P; 210; Mg] counterparts.

The observed thermal properties align well with the wide-angle X-ray diffraction (WAXD) patterns (Figure 5). In both PBS[0T; 1.0P; 210; Mg] and PBS[0T; Ti; 230], we observed fairly sharp and distinct signals at 2θ angles of 19.5, 21.8, and 22.6°. However, these signals became increasingly blurred as BHET units were incorporated into the structure, and the blurring intensified with a higher concentration of BHET units. This blurring effect was marginal in the PBS[*x*T; 1.0P; 210; Mg] series (*x* = 5.0, 10, 15, 20) (Figure 5a). In contrast, the PBS[*x*T; Ti; 230] series exhibited a significant degree of blurring (Figure 5b). This observation was consistent with the ^1^H NMR spectra, specifically, a more random distribution of –O_2_CC_6_H_4_CO_2_– and –OCH_2_CH_2_O– units, and the significant reduction in *T*_m_ values observed in the DSC curves for the PBS[*x*T; Ti; 230] series.

### 3.3. Mechanical Properties

PBS samples with a composition lacking any –O_2_CC_6_H_4_CO_2_– and –OCH_2_CH_2_O– units demonstrated remarkable tensile strength (34 MPa) and elastic modulus (370 MPa). However, the elongation at break exhibited significant variability among specimens with a substantially high average value of 180 ± 130. Upon incorporating BHET units, we observed more consistent tensile data, exhibiting a trend where the yield and ultimate strengths, as well as the elastic modulus, decreased with increasing BHET contents while the elongation at break increased (Figure 6a). The most intriguing sample, PBS[5.0T; 1.0P; 210; Mg], which exhibited a *T*_m_ signal at 105 °C and fast crystallization, demonstrated a substantial yield strength (26 MPa) and elastic modulus (290 MPa), along with an impressive elongation at break (370%). Tensile properties displayed a slight deterioration in the case of PBS[5.0T; 1.0P; 200; Mg], which was synthesized at a lower temperature of 200 °C. In contrast, significantly worse tensile properties were observed for PBS[5.0T; 1.0P; 220; Mg], synthesized at a higher temperature of 220 °C (Table 3, entries 7–8; Appendix A). PBS[5.0T; 2.0P; 210; Mg] and PBS[5.0T; 3.0P; 210; Mg], synthesized with a higher amount of H_3_PO_4_, exhibited a brittle character with low elongation at break values of 8% and 16%, respectively, along with reduced tensile strength of 14 MPa (entries 9–10; Appendix A). Interestingly, the change in metal species did not significantly affect the tensile properties. All three samples, PBS[5.0T; 1.0P; 210; Mg], PBS[5.0T; 1.0P; 210; Zn], and PBS[5.0T; 1.0P; 210; Mn], exhibited similar tensile properties (entries 2 and 11–10; Appendix A). The role of M^2+^ ions is solely to facilitate the connection of two phosphate groups through ionic interaction, and the mechanical properties are not influenced by the variation in M^2+^ ions. It is important to note that metal ions like Mg, Zn, and Mn are essential plant growth nutrients, though the content of Zn ions in soils is strictly regulated below 300 ppm.

PBS[0T; Ti; 230] exhibited a high tensile strength comparable to that of PBS[0T; 1.0P; 210; Mg], although it had a slightly reduced elastic modulus (320 vs. 370 MPa). The elongation at break values showed fluctuations, ranging from 180 to 430%. Upon the incorporation of a small number of BHET units, there was a dramatic improvement in the elongation at break value. PBS[5.0T; Ti; 230] exhibited an elongation at break of 870%, which was significantly higher compared to that of PBS[5.0T; 1.0P; 210; Mg] (870 vs. 370%). However, this improvement came at the expense of reduced yield strength (19 vs. 26 MPa) and a lower elastic modulus (200 vs. 290 MPa). Specimens prepared with PBS[10T; Ti; 230], PBS[15T; Ti; 230], and PBS[20T; Ti; 230] were strong and unbroken, even at the instrument’s ultimate elongation of 1000% (Figure 6b). The yield strength and elastic modulus gradually decreased with the increasing BHET contents. Interestingly, the latter two samples did not exhibit *T*_m_ signals in the second heating scans, but slowly crystallized at room temperature, resulting in initially transparent specimens turning hazy. Tensile properties were measured after this crystallization process.

### 3.4. Rheological Properties

Usually, polymers exhibit shear-thinning behavior in rheology, which means that their viscosity decreases under shear strain. This behavior deviates from Newtonian flow, where viscosity remains constant, regardless of shear strain. Due to the presence of long-chain branches, either through phosphate triester linkages [(–CH_2_O)_3_P=O] or through diester linkages via ionic bonds [(–CH_2_O)_2_P(O)(O^−^)Mg^2+^(O^−^)P(O)], the PBS[*x*T; 1.0P; 210, Mg] (*x* = 0, 5.0, 10, 15, 20) series exhibited significant shear-thinning behavior in the flow curves measured using a rotational rheometer at 170 °C. Specifically, the complex viscosities (η*) of the ionic aggregates PBS[*x*T; 1.0P; 210, Mg] at a low angular frequency of 0.1 rad/s were notably high and varied among the samples, ranging from 8600 to 89000 Pa·s. These values were much higher than the η* value of 5400 Pa·s measured for a commercial PBAT sample with a *M*_w_ of 230 kDa and a Ð of 4.4. As the angular frequency increased, the η* values significantly decreased, with all samples, including PBAT, exhibiting a similar low viscosity level of 250–540 Pa·s at a high angular frequency of 630 rad/s (Figure 7a). This remarkable sensitivity of viscosity to angular frequency is known to be a crucial advantage in various melt processing applications such as blowing and casting films, injection and blow molding, and thermoforming [69,70]. In contrast, the conventional PBSs (PBS[*x*T; Ti; 230]) prepared using conventional methods with the (nBuO)_4_Ti catalyst and lacking long-chain branches did not display such prominent shear-thinning behavior. These conventional PBSs exhibited relatively low η* values of 4100–9500 Pa·s at a low angular frequency of 0.1 rad/s, while at a high angular frequency of 630 rad/s, they showed similar η* values ranging from 510 to 670 Pa·s (Figure 7b).

The cross-over point observed in the storage-modulus and loss-modulus curves measured using a rotational rheometer is correlated with molecular weight and dispersity. Specifically, a lower angular frequency at the crossover point corresponds to a higher molecular weight, while a lower modulus at the crossover point indicates a wider dispersity [71]. In the case of ionic aggregates, such as those prepared in this study by reacting copolyesters containing –CH_2_OP(O)(OH)_2_ and (–CH_2_O)_2_P(O)OH groups with divalent metal 2-ethylhexanoate, it is impossible to measure their molecular weights using GPC analysis. Therefore, estimating their molecular weights and dispersity qualitatively through the crossover points becomes a meaningful approach. For the PBS[*x*T; 1.0P; 210, Mg] (*x* = 0, 5.0, 10, 15, 20) series, these ionic aggregates exhibited crossover points at much lower angular frequencies and lower moduli compared to conventional PBSs (i.e., PBS[*x*T; Ti; 230] series) prepared using conventional methods with (nBuO)_4_Ti catalyst (2–25 vs. 160–320 rad/s; 25–39 vs. 150–170 kPa; Figure 8 and Appendix A). The data suggest that the ionic aggregates, formed by connecting low-molecular-weight chains (*M*_w_ = 44 kDa) with divalent metal ions, behave in the molten state as if they were much higher molecular weight polymers than the conventional ones, which already had significantly high molecular weights according to GPC analysis (*M*_w_ = 200–300 kDa). Furthermore, these findings indicate that the dispersity values of the ionic aggregates are much broader than those of conventional PBS, which aligns with their higher shear-thinning behavior due to the presence of long-chain branches.

### 3.5. Biodegradability

Like PLA, PBS is known to exhibit resistance to degradation under ambient conditions in soil, marine environments, and freshwater, only undergoing composting in harsh industrial composting settings with temperatures reaching 60 °C and an abundance of microbiomes [2]. In a previous study, we investigated the biodegradability of ionic aggregates of PBAT, which demonstrated a significant enhancement in biodegradability when compared to conventional PBAT under ambient soil conditions [58]. Upon that information, biodegradability of the ionic aggregates of PBS (specifically, PBS[5.0T; 1.0P; 210, M], M = Zn, Mg, Mn) were evaluated. Biodegradability assessments were conducted in an ECHO^®^ respirometer (ECHO Instruments, Slovenske Konjice, Slovenia), in accordance with ISO 17556 [66] guidelines, ensuring a constant temperature of 25 °C and a humidity level between 50 and 55%. CO_2_ evolution was continuously monitored under a controlled airflow. During the 110-day incubation period, the ionic aggregates of PBS, containing the same amount of M^2+^, –CH_2_OP(O)(OH)O^−^, (–CH_2_O)_2_P(O)O^−^ groups as in the ionic aggregates of PBAT, did not display a significant improvement in biodegradability. This was evident, as the total CO_2_ evolved from PBS was considerably lower compared to microcrystalline cellulose (Figure 9). The different biodegradability behaviors observed between PBS and PBAT cannot be easily explained. However, it is worth noting that PBAT is known to exhibit superior biodegradability compared to PBS or PLA; certain grades of PBAT obtained certification for biodegradability under ambient soil conditions, whereas the latter two only achieve certification under artificial industrial composting conditions at 58 °C [2]. It is important to recognize that the market’s demand is not necessarily for rapidly biodegradable polymers. What the market seeks are polymers that remain persistent for a certain period during use and subsequently undergo rapid biodegradation.

## 4. Conclusions

A range of PBS-based copolyesters with varying BHET content (5–20 mol% BHET units), specifically containing –O_2_CC_6_H_4_CO_2_– and –OCH_2_CH_2_O– units, were synthesized by SA/[BD + BHET] polycondensation with either 1–3 mol% H_3_PO_4_ or the conventional (nBuO)_4_Ti catalyst. The copolyesters produced using H_3_PO_4_ (PBS[*x*T; *y*P; *T*] series) contained chain-end-positioned monoester (–CH_2_OP(O)(OH)_2_), inner-chain-positioned diester (–CH_2_O)_2_P(O)OH, and branching-point-triester (–CH_2_O)_3_P=O groups. The relative abundance of these groups varied based on reaction temperature and the amount of BHET in the feed. Analysis of ^1^H NMR spectra indicated that –O_2_CC_6_H_4_CO_2_– and –OCH_2_CH_2_O– units were distributed more randomly in the copolyesters prepared using the (nBuO)_4_Ti catalyst. The PBS[*x*T; *y*P; *T*] series were treated with M(2-ethylhexanoate)_2_ (M = Mg, Zn, Mn) in the final stage to create ionic aggregates, PBS[*x*T; *y*P; *T*; *M*] series. These aggregates featured polymer chains extended through ionic bonds between divalent metal cations and –CH_2_OP(O)(OH)O^−^ or (–CH_2_O)_2_P(O)O^−^ anions. DSC studies revealed that the *T*_m_ values gradually decreased with increasing BHET content, with a more significant reduction observed in copolyesters prepared using the (nBuO)_4_Ti catalyst. Specifically, the copolyester containing 5 mol% BHET units, prepared using H_3_PO_4_/Mg(2-ethylhexanoate)_2_ and (nBuO)_4_Ti catalyst, exhibited *T*_m_ values of 105 and 99 °C, respectively. Crystallization rates were also reduced with the addition of BHET units, and only PBS and its copolyesters containing 5 mol% BHET units displayed crystallization signals (*T*_c_) during the cooling scan. Incorporation and increase in BHET units led to a gradual decrease in tensile strength and elastic modulus, while elongation at break increased. Notably, PBS[5.0T; 1.0P; 210; Mg] exhibited impressive mechanical properties with a substantial yield strength (26 MPa) and elastic modulus (290 MPa), along with an elongation at break of 370%. In contrast, its corresponding copolyesters prepared using the (nBuO)_4_Ti catalyst, PBS[5.0T; Ti; 230], displayed a much higher elongation at break of 870%, albeit at the expense of reduced yield strength (19 MPa) and elastic modulus (200 MPa). Due to the presence of long-chain branches, the copolyesters prepared with H_3_PO_4_/Mg(2-ethylhexanoate)_2_ exhibited significant shear-thinning behavior in the flow curves, while the copolyesters prepared using (nBuO)_4_Ti catalysts did not exhibit such prominent shear-thinning behavior. Disappointingly, the ionic aggregates of PBS[5.0T; 1.0P; 210; M] (M = Mg, Zn, Mn) did not display any significant enhancement in biodegradability, in contrast to the rapid biodegradation observed with the ionic aggregates of PBAT.

## 5. Patents

A patent was applied (Kr 10-2023-0175717, 6 December 2023, assigned to Ajou University).

## Data Availability

All data are included in this article and in the Appendix A.

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
