# Peer review of "Ductile Copolyesters Prepared Using Succinic Acid, 1,4-Butanediol, and Bis(2-hydroxyethyl) Terephthalate with Minimizing Generation of Tetrahydrofuran"

_polymers, 2024, doi:10.3390/polym16040519_

Round 1
Reviewer 1 Report
Comments and Suggestions for Authors
This manuscript reports the preparation and characterization of PBS-based copolyesters through the polycondensation of succinic acid (SA) with 12 1,4-butanediol (BD) and bis(2-hydroxyethyl) terephthalate (BHET) and using two different catalysts, H3PO4 and the conventional (nBuO)4Ti.
The topic and the results are very interesting and the characterization is thorough. However, there are some parts to improve in the text which need to clarify.
In particular:
1) The authors had already published a similar synthetic procedure for PBS and now present a copolymerization with a monomer BHET obtained from recycled PET which would require greater attention and comments to evaluate its positive and negative effects. The authors should better underline the expectations that characterize the research even at the beginning of the results by explaining the choice of this monomer
2) It would be appropriate to discuss the role of H3PO4 better because, as analyzed in the results and conclusions, this catalyst also plays a connecting role between different chains linked in the different ester groups (for example in the diester and triester) which is then amplified by the role of the added metal cation to neutralize the remaining acidic positions. The role on PM is then also discussed in terms of concentration (lines 250-269) but it should be highlighted that it is therefore not only a catalytic role but also a connection between chains. It would therefore be appropriate to better define the comparison with the conventional catalytic system, underlining its advantages and disadvantages but taking into consideration the different role of the two compounds mentioned as "catalysts".
3) The evaluation of the ratio between the integrals of the signals in the 31P-NMR spectra shown in fig 1 is an interesting method to describe the species present but a precise evaluation appears difficult due to excessive background noise, especially when varying BHET. Is it possible to evaluate the error of this quantitative evaluation? The authors should perhaps indicate that this is a semi-quantitative assessment which allows for a relative rather than absolute estimate or that below a certain concentration the quantitative evaluation is not possible.
4) In the 1H- NMR discussion: The use of the term “satellite signals” is wrong in my opinion. They are lower intensity signals but not satellites
5) Regarding the analysis of the 1H NMR spectrum of the volatile compounds collected during the esterification: Is the recovery of volatile compounds quantitative? Could there be risks of an undervaluation especially for the most volatile component? Describe how they were collected The following comment on the reduced quantity of THF produced is in my opinion excessive, with reference to previous data and should be better expressed to underline the advantage of using BHET to obtain copolymers already synthesized in the past in a different way. I suggest modifying the text by immediately indicating more directly the advantage obtained compared to the critical issues encountered in the past
6) Line 319: It seems to me there is an error on the last SES indicated in the unit report in the following sentence: ” The ratios of [TET]:[TES]:[SES]:[TBT]:[TBS]:[SES] measured in 1H NMR spectra were notably different from those observed for copolyesters prepared using H3PO4”.
7) Thermal properties: Regarding the similar thermal behavior of PBS obtained in the absence of BHET with the two different catalytic systems: how can this result be justified considering the differences observed for the other structural characteristics? PM, presence of ionic bonds etc
In conclusion, in my opinion, the authors should carefully review the manuscript and several corrections are suggested for the publication on Polymers.
Reviewer 2 Report
Comments and Suggestions for Authors
This manuscript aims to introduce a new synthesis routine to obtain PBS-based copolyesters. The authors provide detailed description of their synthesis methods and present comprehensive characterization of the materials obtained from their method. Based on their results, the copolyesters produced by their method show somewhat improved mechanical properties and shear-thinning behavior compared with copolyesters synthesized with conventional catalyst. I think overall this manuscript is robust in terms of experimental design and results presentation. However, I have some concerns over its novelty, especially compared to their previous article ‘Rapid Biodegradable Ionic Aggregates of Polyesters Constructed with Fertilizer Ingredients’. Thus, I cannot recommend the publication of this manuscript until the authors address my concerns.
1. The novelty of this manuscript. I think this paper does not convey the importance of this research and what the methodology proposed in the paper could provide to the development of the field. In particular, I think there are some places where the authors could expand more to help readers understand the importance of their research:
a. The title is too vague. It only states the synthesis method and has no information about the advantages or applications of copolyesters obtained by this method.
b. The abstract and conclusions merely repeat the properties of the materials which are already discussed throughout the manuscript. I was wondering if the authors could spend more time discussing what those properties mean for the applications of this material and what are the advantages of their method.
c. The authors only state that the new material does not have improved biodegradability which seems to significantly undermine the importance of this manuscript since the authors put a lot of emphasis on the sustainability issue. While this is what it is, I was wondering if the authors could at least discuss more on why this material does not have good biodegradability and what can be done to potentially improve it.
d. I was wondering if the authors could explain more what the importance of this manuscript compared with ‘Rapid Biodegradable Ionic Aggregates of Polyesters Constructed with Fertilizer Ingredients’ in particular. It seems to be just another application of the same method and the resulted materials also do not exhibit exceptional properties.
2. In section 3.3, the authors mentioned the change of metal species does not alter the tensile properties of the materials much. I was wondering if the authors could comment more on the reasons behind this and what is the main purpose of using different ions in this study.
3. In Figure 7, the flow curves for different PBS[xT; 1.0P; 210, Mg] do not shift up or down monotonically with the increase of x. I was wondering if the authors could comment more on why the materials exhibit such a rheological behavior.
Round 2
Reviewer 1 Report
Comments and Suggestions for Authors
The comments were accepted by the authors by including the requested revisions
Reviewer 2 Report
Comments and Suggestions for Authors
I think the authors have addressed the major concerns I had. Thus, I recommend the publication of this manuscript.